

# Measurement of geologic nitrogen using mass spectrometry, colourimetry, and a newly adapted fluorometry technique

Benjamin W. Johnson[1], Natashia Drage[1], Jody Spence[1], Nova Hanson[2], Rana El-Sabaawi[2], and Colin Goldblatt[1]

[1]School of Earth and Ocean Sciences, University of Victoria, Victoria, BC Canada
[2]Department of Biology, University of Victoria, Victoria, BC, Canada

*Correspondence to:* Benjamin W. Johnson (bwjohnso@uvic.ca)

**Abstract.** Long viewed as a mostly noble, atmospheric species, recent work demonstrates that nitrogen in fact cycles throughout the Earth system, including the atmosphere, biosphere, oceans, and solid Earth. Despite this new-found behaviour, more thorough investigation of N in geologic materials is limited due to its low concentration (1 to 10s ppm) and difficulty in analysis. In addition, N can exist in multiple species ($NO_3^-$, $NH_4^+$, $N_2$, organic-N), and determining which species is actually quantified

can be difficult. In rocks and minerals, $NH_4^+$ is the most stable form of N over geologic time scales. As such, techniques designed to measure $NH_4^+$ can be particularly useful.

We measured a number of geochemical rock standards using three different techniques: mass spectrometry, colourimetry, and fluorometry. The fluorometry approach is a novel adaptation of a technique commonly used in biologic science, applied herein to geologic $NH_4^+$. Briefly, $NH_4^+$ can be quantified by HF-dissolution, neutralization, addition of a fluorescing reagent,

and analysis on a standard fluorometer. We reproduce published values for several rock standards (BCR-2, BHVO-2, and G-2), especially if an additional distillation step is performed. While it is difficult to assess quality of each method, due to lack of international geologic N standards, fluorometry appears better suited to analyzing mineral-bound $NH_4^+$ than mass spectrometry, and is a simpler, quicker alternative to colourimetry.

To demonstrate a potential application of fluorometry, we calculated a continental crust N budget based on new measure-

ments. We used glacial tills as a proxy for upper crust and analyzed several poorly constrained rock types (volcanics, mid-crustal xenoliths) to determine that the continental crust contains $\sim 2 \times 10^{18}$ kg N. This estimate is consistent with recent budget estimates, and shows that fluorometry is appropriate for large-scale questions where high sample throughput is helpful.

Lastly, we report the first $\delta^{15}N$ values of six rock standards: BCR-2 ($1.05 \pm 0.4$‰), BHVO-2 ($-0.3 \pm 0.2$‰), G-2 ($1.23 \pm 1.32$‰), LKSD-4 ($3.59 \pm 0.1$‰), Till-4 ($6.33 \pm 0.1$‰), and SY-4 ($2.13 \pm 0.5$‰). The need for international geologic N standards

is crucial for further investigation of the Earth system N cycle, and we suggest that existing rock standards may be suited to this need.





## 1 Introduction

Since its classification as an atmophile element by Goldschmidt (1937), the fate and nature of N in rocks and minerals has received little attention. Many early budgets suggested most of Earth's N was assumed to be in the atmosphere, with only minor amounts in the biosphere, crust, and mantle (e.g., Wlotzka, 1972). And while concentrations are low, typically 1 to 10s

of ppm, the great mass of the solid earth compared to the atmosphere allows for a substantial amount of planetary N to be in the Bulk Silicate Earth (BSE). In fact, the BSE and core likely contain the majority of N in the Earth (Johnson and Goldblatt, 2015). In addition, enriched $\delta^{15}$N values from mantle-derived rocks and the correlation of $N_2$ with $^{40}$Ar indicates that N has cycled between the surface and the deeper planet over geologic time (Marty, 1995; Busigny et al., 2011; Barry and Hilton, 2016).

In spite of the new-found richness of the geologic N cycle, the relative paucity of sample analyses limits robust interpretation or modelling of N cycling over Earth's history. This paucity is due in large part to the difficulty of measuring low concentrations of N in rocks and minerals. Though a variety of analytical techniques are now able to measure N at ppm-level concentrations in rocks and minerals (e.g., Bräuer and Hahne, 2005; Barry et al., 2012; von der Handt and Dalou, 2016), several of these are either analytically expensive or only operational at a handful of labs around the world. The development of techniques

that are more easily accessible and able to be performed with standard geochemical equipment would be a great benefit to the community.

In this study, we adapt a fluorometry technique developed by Holmes et al. (1999) that is commonly used in biologic and aquatic chemistry studies, and compare it with two other techniques: colourimetry (Hall, 1993) and elemental analyzer mass spectrometry (e.g., Stüeken, 2013). Through analysis of a number of rock standards, we demonstrate that, while this fluorometry

technique has some associated uncertainty, it reproduces published values for standards BCR-2, BHVO-2, and G-2, especially if a distillation step is undertaken. It also performs better than elemental analyzer combustion mass spectrometry or colourimetry methods for quantifying N in crystalline rocks. The fluorometry technique has the advantage over other techniques by being relatively fast and straight forward, as well as specifically targeting $NH_4^+$. There are three main benefits: the potential to increase the number of analyses of N in geologic samples due to relative ease of the method; a screening method that can be

used to guide further isotopic investigation; as it only measures $NH_4^+$, this technique can also be helpful in determining which phase of N is found in geologic samples, which can be difficult using other techniques.

In addition, we also present a preliminary application of the method vis-á-vis a N budget for the continental crust based on glacial tills and crystalline crustal rocks from North America. Along with the atmosphere and mantle, the continental crust is one of the main N reservoirs on the planet (Goldblatt et al., 2009; Johnson and Goldblatt, 2015), thus determining its content

is key in the evolution of the N cycle over time.

We also call for the development international geologic N standards (after Ader et al., 2016). Method development for measurement of geologic N suffers without such standards. We present the first $\delta^{15}$N values of a number of rock standards (BCR-2, BHVO-2, SY-4, LKSD-4, Till-4, G-2), and suggest they may be suited for geologic N standards, given more thorough analysis.



**Table 1.** Rock standards from United States Geological Survey (USGS) and Geological Survey of Canada (GSC) analyzed with published values, if any, and N analysis reference.

| Standard | Description | N (ppm) | Reference |
|---|---|---|---|
| BCR-1/2 | USGS Columbia river basalt | $34 \pm 12$ | 1, 2, 3 |
| BHVO-2 | USGS Hawaiian basalt | $22.6 \pm 3$ | 1, 3 |
| G-2 | USGS Paleozoic granite | $34 \pm 4$ | 1, 3 |
| Till-4 | GSC Till from Scisson's Brook, New Brunswick | | |
| LKSD-4 | Big Gull Lake sediment, Ontario | | |
| SY-4 | Diorite Gneiss, Ontario | | |

[1]Govindaraju (1994),[2]Norris and Schaeffer (1982), [3]Murty et al. (1983)                                                                          ,

## 2 Methods

### 2.1 Rock standards and samples

We analyzed a number of geochemical rock standards (Table 1). Several standards (LKSD-4, Till-4, SY-4) have no previous N concentration measurements, to our knowledge. Remaining standards have published N concentrations (BCR-2, BHVO-2, G-2), with values reported from neutron activation analysis (NAA). This technique works by irradiating samples with neutrons to transform $^{14}$N into $^{14}$C, where the resulting material can then be purified and assayed radiochemically as a proxy for N concentration (Shukla et al., 1978).

In addition to rock standards, we analyzed a number of other lithologies, including glacial tills, basalts, granites, and carbonates. These samples were chosen as a proof of concept for the adapted fluorometry technique, namely, investigating the N budget of the continental crust. All tills are from British Columbia, Canada, and have eroded a variety of Phanerozoic crustal lithologies. Basalts and welded tuffs are from the Bonanza Arc and Sicker Group on Vancouver Island, British Columbia. Granites come from a variety of locations in North America. Sample descriptions are given in the supplemental material.

### 2.2 Rock Sample Preparation

Using a rock saw, rock samples were cut into small blocks and the weathered edges were removed. Rock powders were prepared by crushing the blocks in a steel jaw crusher and then powdered using a shatterbox with a tungsten carbide puck. The shatterbox puck and chamber were cleaned in between each crushing step using deionized water and ethanol. Clean quartz sand was run between each sample to prevent cross-contamination.



## 2.3 Method 1: Elemental analyzer mass spectrometry

We analyzed all rock standards at the University of Washington Isolab facilities following Stüeken (2013). First, $\sim 1$ g of each sample was weighed into a clean (baked at $500\,^{\circ}$C overnight) Pyrex test tube. Then, $\sim 10$ mL 6N HCl was added, stirred with a glass stir rod, sonicated for 30 minutes, and left in an oven set to $60\,^{\circ}$C overnight. Tubes were then centrifuged to settle

suspended sediment, acid was decanted, fresh acid was added, and the samples were sonicated and dried a second time as above. This decarbonation was done once more. Subsequently, we rinsed samples 3 times in DI-$H_2$O, and all were dried for 3 days at $60\,^{\circ}$C.

Between 12 to 150 mg of decarbonated sample in 9x5 mm Sn capsules was flash combusted at $1000\,^{\circ}$C in a Costech ECS 4010 Elemental Analyzer with an excess of $O_2$. Combustion products passed over a reduced copper column at $650\,^{\circ}$C to reduce

all N to $N_2$ and absorb excess $O_2$ and a magnesium perchlorate trap to absorb water. Sample gas then passed through a 3 m gas chromatography column to separate $N_2$ from $CO_2$ before being analyzed on a Finnigan MAT253 continuous flow isotope-ratio mass spectrometer via a ThermoFinnigan Conflo III. All analyses were quantified using IsoDat software. Errors reported are standard deviations from repeated analyses. We used the following isotopic standards: two glutamic acids (GA-1 and GA-2), dried salmon (SA), and an internal rock standard (McRae Shale).

## 2.4 Method 2: Colourimetric

We followed the procedure outlined in Hall (1993) to analyze samples using colourimetry.

### 2.4.1 Reagent list

*KOH*: A 25% mass:volume solution of KOH was used for HF-neutralizing. We dissolved 250 g KOH in 1 L of water.

*Phenol reagent*: We weighed out 7.0 g of crystalline phenol and 0.02 g sodium nitroprusside into a 200 mL beaker. To this,

we added 20 mL of KOH reagent and 60 mL of deionized water. Solids were stirred to dissolve, and then the solution was topped up to 100 mL total volume with DI-$H_2$O.

*NaOCl*: We diluted 20 mL of commercially available NaOCl to 100 mL total volume with DI-$H_2$O.

### 2.4.2 Stock Solution and working standards

A 1 g $NH_4^+$/L stock solution was prepared in a 250 mL volumetric flask by dissolving 0.7433 g of $NH_4^+$Cl salt in 250 mL DI-

$H_2$O. From this, a secondary ammonium standard solution of 0.2 g $NH_4^+$/L was prepared in a 100 mL volumetric flask using 20 mL of the ammonium stock solution, topped with DI-$H_2$O. Stock solution was diluted to make working standard solutions of 0.005, 0.01, 0.05, 0.1, and 0.2 g/L, which were used to construct standard curves for sample concentration determination.

### 2.4.3 Sample Digestion

Working standards, blanks (DI$-H_2$O), and rock powder were subjected to HF treatment at room temperature in 25 mL Teflon

vials. To these vials, we added 2 mL of working standard, 2 mL of blank, or approximately 0.25 g of rock powder. The samples



were digested for seven days in 2 mL of 50% hydrofluoric acid in a fume hood. We swirled vials every two days to facilitate digestion.

After sample digestion period, the solution was neutralized by adding 20 mL of 25% KOH to each vial. Exploratory analysis of different aliquots from the top, middle, and bottom of each vial gave different results, thus the solution was stirred with a glass rod to ensure homogenization. The stir-rod was rinsed with DI-H$_2$O in between samples to prevent cross-contamination. Homogenized solutions sat for 15 minutes to allow suspended rock powder to settle.

### 2.4.4 Colourimetric analysis

All liquid, plus undissolved solids, were placed in a 100 mL round bottom flask, which was attached to a standard distillation setup. Samples were boiled for $\sim$ 10 to 15 minutes, and 8 mL of distillate was collected into 8 mL of 0.01 N H$_2$SO$_4$. To this, 1 mL of phenol reagent, 1 mL of NaOCl, and 5 mL DI-H$_2$O were added and stirred. The colour-change reaction was allowed to proceed for 2 hours. Absorbance was measured at 630 nm in a plastic cuvette on an Ocean Optics spectrophotometer and quantified using SpectraSuite software.

Standard solutions were processed the same way as the samples. We made a standard curve of absorbance plotted against starting concentration (0.005, 0.01, 0.05, 0.1, and 0.2 g/L) of standard solutions. This was used to calculate sample concentrations, applying appropriate dilution corrections.

## 2.5 Method 3: Fluorometric

The following is a detailed description of our adaptation of the Holmes et al. (1999) method. Key considerations and suggestions for improvement are discussed in the following section.

### 2.5.1 Reagent list

*Working reagent (WR)*: We made a mixture of sodium sulphite, borate buffer and orthophthaldialdehyde (OPA) solutions. The procedure for preparing the working reagent follows Holmes et al. (1999). To make the sodium sulfite solution, 0.2 g of sodium sulfite was added to 25 mL of DI-H$_2$O. For the borate buffer, 80 g of sodium tetraborate was added to 2 L of DI-H$_2$O, which was then stirred for 4 hours with a stir bar. To make the OPA solution, 4 g of OPA was added to 100 mL of 95% ethanol and protected from the light while stirred with a stir bar. The borate buffer, OPA solution and 10mL of the sodium sulfite were mixed in a $>$ 2 L brown polyethylene bottle. The working reagent mixture sat for at least one day prior to use.

### 2.5.2 Stock Solution and working standards

The same stock solution was used for fluorometry as for colourimetry. A range of working standards were made by diluting stock solution (0.2 g NH$_4{}^+$/L) into reaction bottles, resulting in concentrations of 0.005, 0.01, 0.05, and 0.1 g NH$_4{}^+$/L. This range of working standards was used to construct standard curves.



### 2.5.3 Sample digestion

The same digestion and neutralization procedures were used for fluorometry as for colourimetry.

### 2.5.4 Optional distillation

Some replicates of samples were distilled, as described in section 2.4. As this step is the most time-intensive step aside from
digestion, we ran most analyses without distilling. As discussed later, this step may be useful for samples with either low N or
samples that are difficult to digest completely (e.g., G2).

### 2.5.5 Fluorometric analysis

Brown 50 mL polyethylene bottles were used for the fluorescing reaction. The reaction bottles were first emptied of their
storage solution (i.e., clean working reagent) and rinsed with 5 mL DI-$H_2O$. Using a pipette, 10mL of DI-$H_2O$ was added
to each reaction bottle. Subsequently, an aliquot of the neutralized solution was added. About 0.1 mL of sample solution was
added to each bottle. Then, 2.5 mL of working reagent was added to each reaction bottle. After adding the working reagent,
the reaction bottle was inverted to homogenize.

Immediately after homogenization, an aliquot of solution from the reaction bottle was transferred to a plastic cuvette, and
fluorescence was measured in a 1 cm plastic cuvette using a Turner Designs AquaFluor fluorometer. After this initial measure-
ment, the aliquot of solution was transferred back into its reaction bottle, the cuvette was rinsed, and the bottle was capped and
inverted several times to homogenize. The sample then reacted for three hours. After three hours, fluorescence was measured
in three sample aliquots. Sample concentrations were calculated using a standard curve (Fig. 1).

After measurement, the remaining solution in the reaction bottle was emptied and a small amount ( 2 mL each) of working
reagent and DI-$H_2O$ was added to the bottles for storage.

### 2.5.6 Calculation of sample $NH_4$ concentration

Net fluorescence ($F_{net}$) for working standards and samples were calculated by averaging the three final fluorescence ($\bar{F}$) read-
ings, subtracting initial fluorescence ($Fi$), then subtracting average fluorescence minus initial from the blank ($\bar{B}$) (Eq. 1).

$$F_{net} = \bar{F} - Fi - \bar{B} \tag{1}$$

Then, using standard curves (either corrected to digestion vials or in reaction bottles, slope = $s$) sample concentration in either
digestion vials or reaction bottles was calculated. Importantly, we chose to force standard curves through the origin.

Final $NH_4^+$ concentrations ($[NH_4^+]$) were calculated by correcting concentration to KOH-neutralized vial, multiplying con-
centration by total volume in vial ($D$) to determine a mass of $NH_4^+$. Then, we divided this $NH_4^+$ mass by starting sample rock
powder mass ($m$) to get the concentration of $NH_4^+$ in ppm-mass (Eq. 2).

$$[NH_4^+] = \frac{F_{net}}{s} \frac{D}{m} \tag{2}$$





**Table 2.** Nitrogen and $\delta^{15}$N data from colourimetric and mass spectrometry analyses. Both techniques appear to underestimate N concentration, perhaps due to incomplete N-extraction during combustion for mass spectrometry or incomplete recovery of $NH_4^+$ distillation. Concentrations are in ppm.

| Standard | Published | Colourimetric | Mass spectrometry | $\delta^{15}$N |
|---|---|---|---|---|
| BCR-1/2 | $34 \pm 12$ | $12.6 \pm 8$ | 21 | $1.05 \pm 0.4$ |
| BHVO-2 | $22.6 \pm 3$ | $3.5 \pm 0.7$ | $13.3 \pm 0.6$ | $-0.3 \pm 0.2$ |
| G-2 | $34 \pm 4$ | $1.6 \pm 0.9$ | $5 \pm 0.7$ | $1.23 \pm 1.32$ |
| SY-4 | | $6.9 \pm 2.8$ | $14.3 \pm 0.6$ | $2.13 \pm 0.5$ |
| LKSD-4 | | $487 \pm 401$ | $16000 \pm 8$ | $3.59 \pm 0.1$ |
| Till-4 | | $82.2 \pm 40$ | $440 \pm 2$ | $6.33 \pm 0.1$ |
| Carb | | 66.1 | $48.5 \pm 1.3$ | |

## 3 Results

### 3.1 Method 1: Mass spectrometry

Elemental analyzer mass spectrometry was able to measure N concentration in all rock standards (Table 2). Analyses for crystalline standards are lower than published values, likely due to the incomplete liberation of N from silicate lattices during combustion (Bräuer and Hahne, 2005). Values for BCR-1/2 are 62% of published, BHVO-2 59% of published, and G-2 15% of published. Values reported herein for Till-4, SY-4, and LKSD-4 are the first to our knowledge.

In addition, we report $\delta^{15}$N values for all samples (Table 2). While more analyses would need to occur for these to be used as isotopic standards, the fact that there is measurable N in all standards suggests they may be suitable candidates for geologic N-isotopic standards. Samples with high N (presumably some organic N), LKSD-4 and Till-4, could be ideal.

### 3.2 Method 2: Colourimetric

The colourimetric method was able to analyze all rock standards (Table 2), with standard curves having r$^2$ values above 0.99 (Fig 1). Measured concentrations are lower than published values, at 37% , 15% and 5% of published values for BCR-2 , BHVO-2, and G-2, respectively (Table 2). We suggest that this potential underestimate is due, in part, to difficulty in quantitatively distilling all $NH_4^+$ from a dissolved sample. The distillation apparatus is imprecise in nature, and controlling the final volume of distillate is difficult.




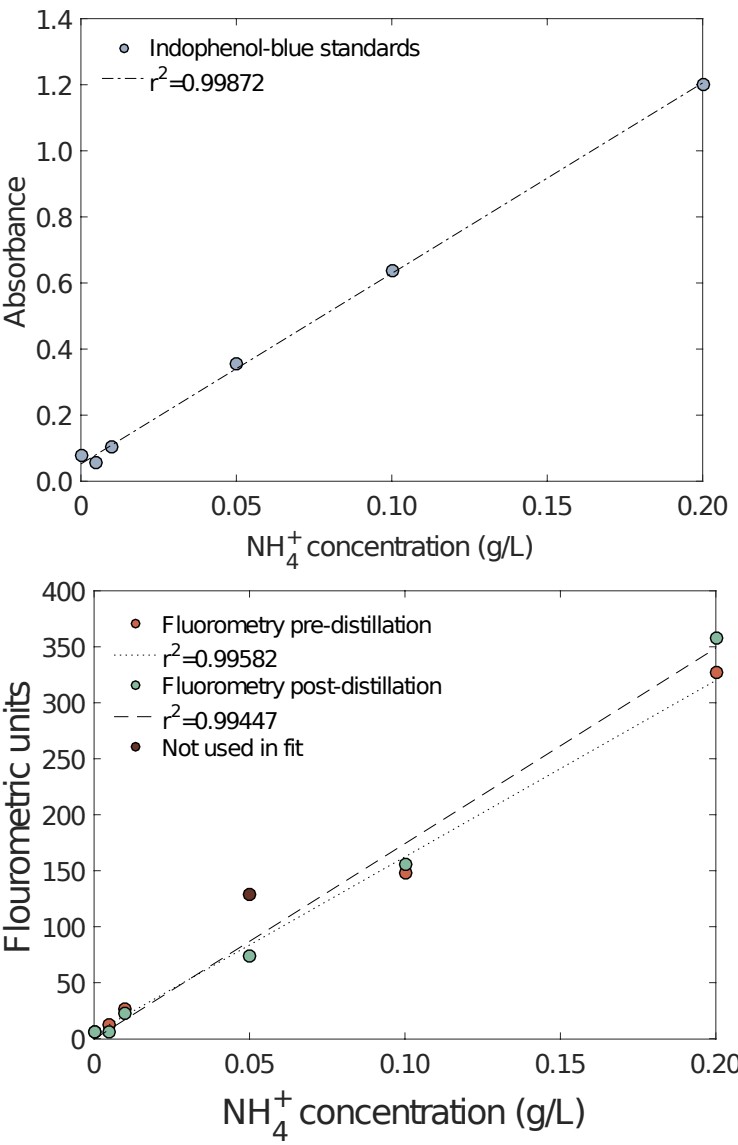

**Figure 1.** Example standard curves for colourimetry and fluorometry analyses. These standard curves were used to calculate sample values for those analyses occurring after distillation. In addition, fluorometry curves show that distillation does not greatly affect working standards during distillation, though the same may not be true for actual samples.



**Table 3.** Averages and standard deviations of repeated rock standard analyses using the fluorometry technique, with and without distillation, compared with published values. Number of measurements (NOM) indicates the number separate measurements for "no distillation" samples. All concentrations are in ppm.

| Standard | Published | post-distillation | no distillation | NOM |
|---|---|---|---|---|
| BCR-1/2 | $34 \pm 12$ | $38 \pm 1$ | $33 \pm 8.3$ | 36 |
| BHVO-2 | $22.6 \pm 3$ | $35 \pm 1$ | $15 \pm 5.7$ | 9 |
| G-2 | $34 \pm 4$ | $36 \pm 4$ | $11 \pm 4.9$ | 13 |
| SY-4 | | $11 \pm 1$ | $5.2 \pm 4.5$ | 9 |
| LKSD-4 | | $9300 \pm 2150$ | $5200 \pm 1000$ | 9 |
| Till-4 | | $455 \pm 36$ | $71 \pm 25$ | 15 |
| Carb | | 37 | $93 \pm 18$ | 12 |

### 3.3 Method 3: Fluorometric method

### 3.4 Rock standards

Results for rock standard analyses are shown in Table 3. Values range from 5.2 to 5200 ppm by mass. Analyses without distillation match published values within error for BCR-2 and BHVO-2, with values of $33 \pm 8.3$ and $15 \pm 5.7$ ppm, compared to $34 \pm 12$ and $22.6 \pm 3$, respectively. Analyses of G-2 are lower than published, at $11 \pm 4.9$ from our analyses and $34 \pm 4$ from literature values. Other standards have values of $5.2 \pm 4.5$ (SY-4), $5200 \pm 1000$ (LKSD-4), and $71 \pm 25$ ppm. Analyses after distillation are generally higher than those with no distillation step.

### 3.5 Continental Crust

Measured $NH_4^+$ concentrations from a number of continental crust samples are shown in Table 4. We have included analyses of BCR-2 in the "volcanics" category. In general samples of all crystalline rock types have between 10 to 40 ppm, and sedimentary rocks are higher, with values between 100 and 1000 ppm.



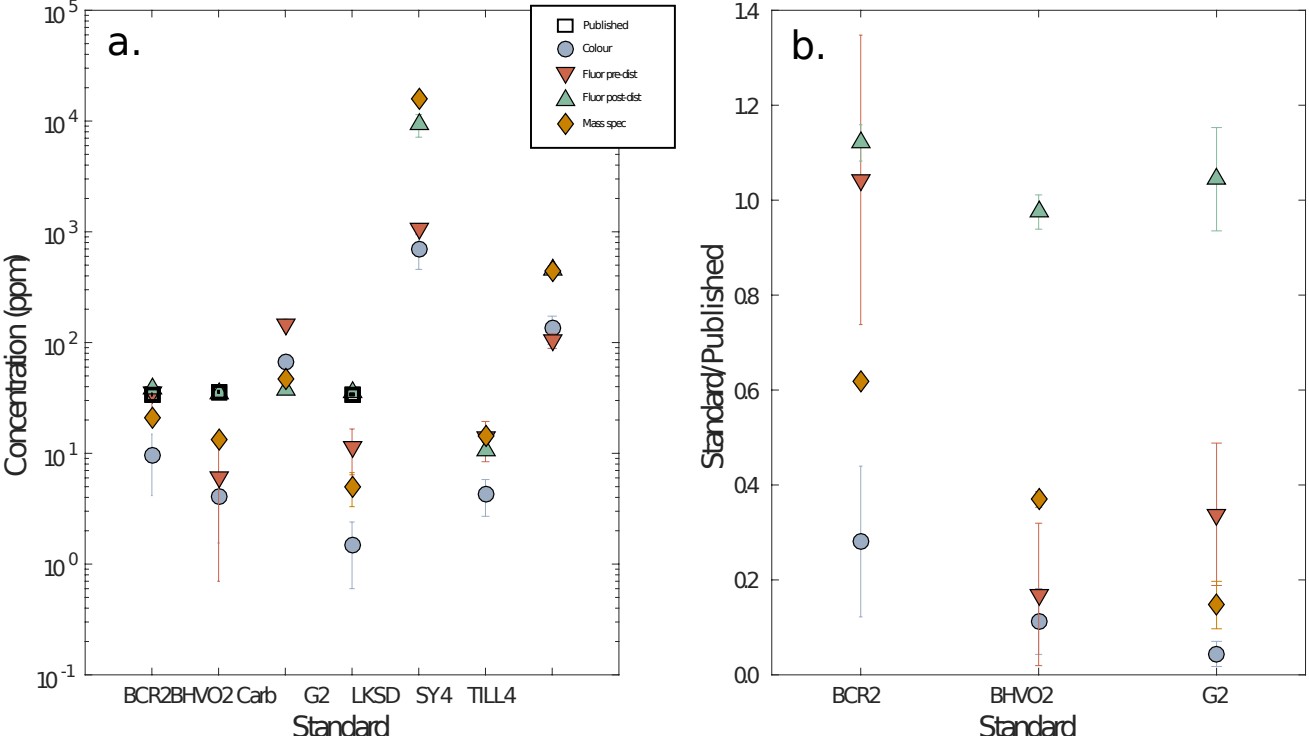

**Figure 2.** a. Comparison of N concentrations in rock standards from three different methods. colourimetry consistently yields lower values than fluorometry, and values typically lower than EA-mass spectrometry. Fluorometry and EA-mass spectrometry give similar results for sedimentary rocks (LKSD-4, Till-4, 5.1), but tend to differ for crystalline rocks (BCR-2, BHVO-2, SY-4, G-2), with fluorometry giving higher concentrations. b. Though associated errors are larger, fluorometry does reproduce published N concentration values from three standards (BCR-2, BHVO-2, G-2), especially after distillation.

# 4 Discussion

## 4.1 Fluorometry

### 4.1.1 Reproducibility and reproduction of published values

Reproducibility of repeated analysis was between $10 - 50\%$ of mean values for all rock standards (Table 3). Overall, error appears to be lowest for samples which are most easily digested: BCR-2 and carbonate. More felsic standards, such as G-2, have higher error but also show lower concentrations than previous work. On runs with distillation, G-2 ($36 \pm 4$ ppm) matched published values ($34 \pm 4$ ppm), BCR-1/2 ($38 \pm 1$ ppm) also matched published values ($34 \pm 12$ ppm), and BHVO-2 ($35 \pm 1$ ppm) had higher than published values ($22.6 \pm 3$ ppm). Runs without distillation gave concentrations below their published values,



**Table 4.** Nitrogen concentration in upper crustal rocks using the fluorometry method.

| Rock type | Mean | Standard deviation | No. samples |
|---|---|---|---|
| Tills | 81.2 | 36.4 | 8 |
| Silt | 1060 | 113 | 1 |
| Volcanics | 21.4 | 12.5 | 13 |
| Carbonates | 114.2 | 40.9 | 1 |
| Granitic | 30.8 | 31.2 | 5 |
| Gabbro | 11.3 | 12.6 | 5 |
| Gneiss | 29.8 | 0.8 | 1 |
| Xenolith | 34.4 | 16.1 | 8 |

with $11 \pm 4.9$ ppm for G-2 and $15 \pm 5.7$ BHVO-2. Therefore, distillation appears to improve agreement between fluorometry and NAA, though this may be coincidence, as NAA analysis have several unresolved issues with calculation accuracy.

We stress that although NAA is of appropriate sensitivity to measure ppm-level N concentrations (e.g., Kolesov, 1995), there are complicating issues. Neutron bombardment can also create $^{14}C$ via reaction with $^{17}O$, with theoretical apparent N

contribution in a sample with $40\%$ O of 18 ppm, though analysis of a synthetic $Al_2O_3$ doped with $20\%$ $^{17}O$ suggests the actual effect may only be 6 ppm (Shukla et al., 1978). In addition, 52 analyses of BCR-1 by Murty et al. (1983) yielded a range in concentration from 15 to 62 ppm. The authors suggest this is due to heterogeneous distribution of N in BCR-1. Alternately, such a range in concentration could be due to adsorption of atmospheric $N_2$. Though sample powders are prepared for irradiation after vacuum pumping (or heating), the possibility of atmospheric $N_2$ contamination appears unresolved Norris and Schaeffer

(1982). Thus, though NAA-analyses of N in rock standards should be able to quantify total N in a sample, several outstanding issues prevent concentrations reported in the literature from being accepted as geochemical standards. A major difficulty in the development of new techniques for measuring geologic N is a lack of international standards.

### 4.1.2   Effects of KOH

The most significant parameter affecting the quality of the standard curves was the concentration of KOH in the reaction

bottles. For a given concentration of $NH_4^+$, varying KOH concentration altered resulting fluorometry readings (Fig. 3). Since the fluorescing reaction is pH dependent (operating between pH 8-10 due to borate buffer), addition of excess KOH causes the pH of the solutions to be too high, and the fluorescing reaction to be inhibited. We adjusted sample dilutions and volumes to result in KOH concentrations in reaction bottles to be between 0.05 to $0.2\%$.

We have verified that small KOH concentrations, as described above, do not affect standard curves (Fig. 4). Curves prepared

on multiple days with water were identical to those produced with KOH, indicating that the fluorescing reaction had gone to




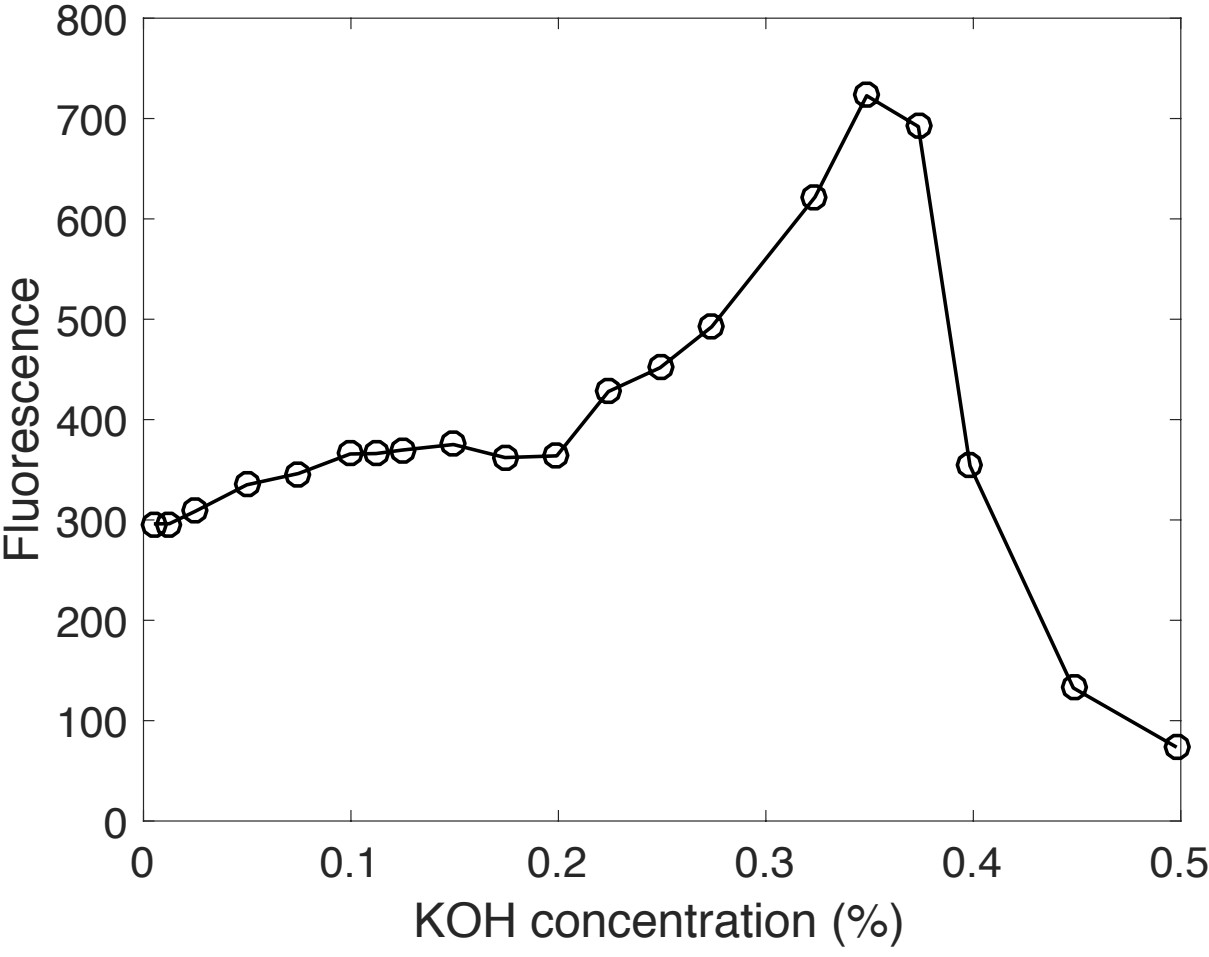

**Figure 3.** Sensitivity test for samples with 0.010 g/L $NH_4^+$ and varying concentrations of KOH in reaction bottles. KOH clearly has an effect on fluorescence. Fluorescence values are flat between 0 to 0.2% KOH, with a large increase approaching 0.35% KOH, and a steep drop off at higher KOH concentrations. We suggest that high concentrations of KOH overwhelm the buffering capacity of the working reagent and inhibit completion of the fluorescing reaction. All reported runs herein had between 0.05 to 0.2% KOH. Controlling KOH concentration is a key factor in the success of this method.





**Figure 4.** Standard curves from three different days comparing preparations made with KOH-solutions to those with only water. Since curves from the same day are indistinguishable with and without KOH, the fluorescing reaction is not affected by the presence of KOH (see text for details). Some variability exists between runs, especially at lower concentrations, but variations are small compared to changes in fluorescence units with changing concentration.

completion. High corrected fluorescence values were due to corrections for dilutions occurring when KOH was added to the initial standard volume, as well as dilutions occurring when preparing reaction bottles. Though there is some variability at concentrations below 0.020 g/L from day to day, similarity of KOH-curves and water only curves implies that the standard curve technique is viable.





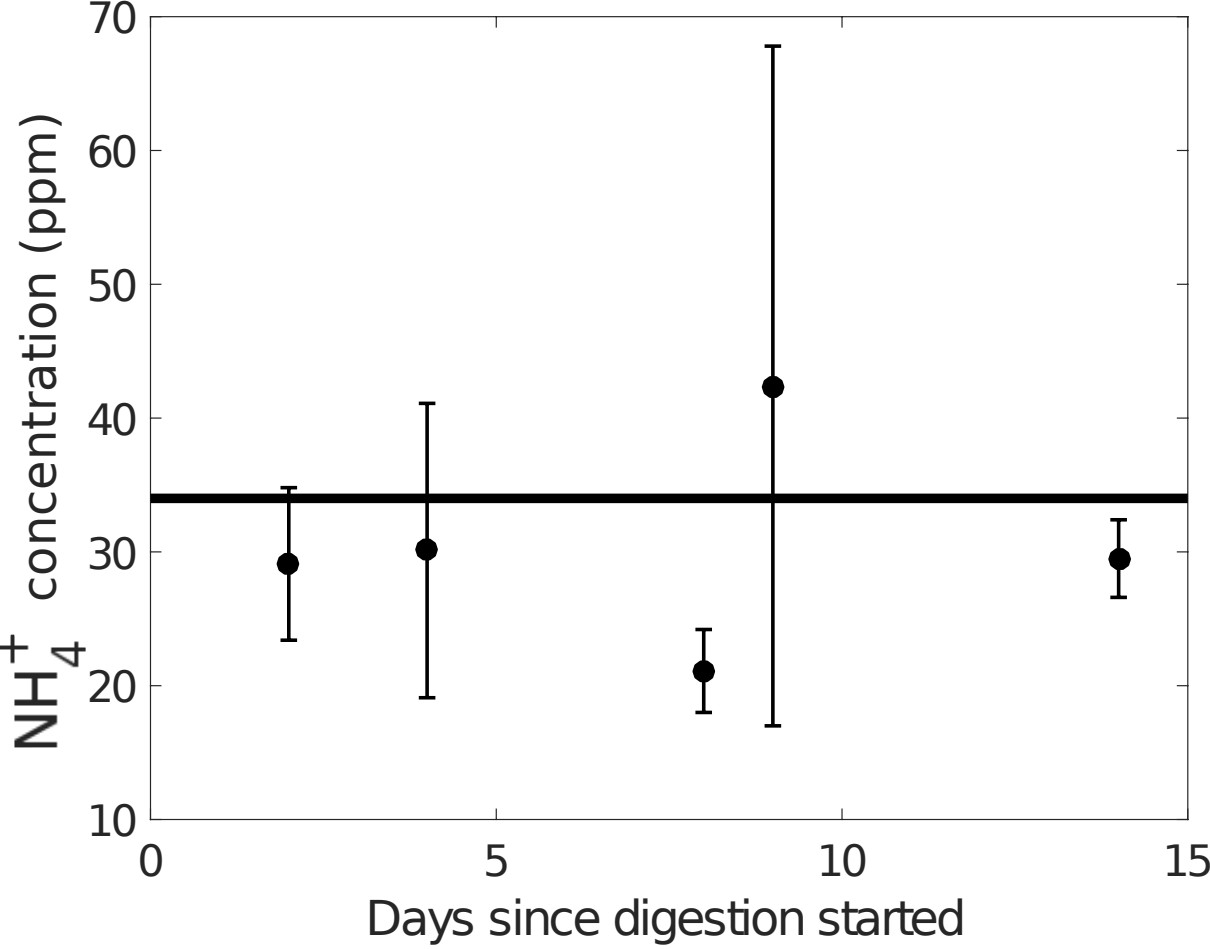

**Figure 5.** Digestion length test for BCR-2 by the fluorometry technique. Samples were digested in 2 mL 50% HF for the number of days indicated. Analyses reproduce published value (34 ppm, black line) within error after only 2 days, and, aside from one analysis 8 days after digestion began, approximate this value afterwards. There is large error in some samples compared to Table 3, likely due to increased number of analyses reported in Table 3, decreasing standard deviation.

### 4.1.3 Digestion length

As sample digestion is only partial, it is possible that length of digestion has an effect on final concentration readings. We conducted a digestion length test for BCR-2 (Fig. 5), and found no clear relationship between digestion length and calculated N concentration. It appears as though there are other factors that have a greater effect on N concentration. Determining the

5  length of time needed for the extraction of all $NH_4^+$ would be an important step, as this could increase sample processing efficiency and sample throughput.





## 4.2 Methods comparison: pros and cons

The main difference between methods discussed in this work is that two (mass spectrometry and NAA) analyze total N while colourimetry and fluorometry specifically target $NH_4^+$. While assaying total N may be advantageous in samples with mixed N speciation (e.g., sedimentary rocks), total N analysis has a more difficult time accounting for $N_2$-contamination from the atmosphere. As rock samples are commonly ground to a fine powder before analysis, the possibility of adsorbing some $N_2$ could affect the accuracy of total N methods, especially where mineral-bound N concentrations are low. Targeted analysis of $NH_4^+$ is likely more better suited to crystalline rocks, where $NH_4^+$ is the primary N-species.

Given the expense of installing a mass spectrometer, the relative inaccessibility of neutron irradiation sources, and the time required for NAA (weeks to months), fluorometry presents a relatively quick and straight forward alternative. All equipment and reagents are easily obtainable. Fluorometers are much more affordable ($3,000 to $10,000) than mass spectrometers ($100,000s), and do not require any supporting infrastructure, and maintenance is comparatively low. And while the technique in its current state may not fully liberate $NH_4^+$ in all samples, it performs with similar reproducibility to mass spectrometry and NAA. Fluorometry also performs well without distillation, required for colourimetry, which takes 15-20 minutes per sample, limiting throughput. Additionally, it is difficult to consistently distill the same volume for each sample, which limits accuracy and reproducibility.

In regards to fluorometric reagents needed, while HF is extremely dangerous, it is more commonly used in geochemistry than any of the colourimetric reagents (especially phenol and sodium nitroprusside), and with appropriate training and caution can be handled safely. Fluorometric reagents are also less hazardous than colourimetric reagents, and are linked with fewer long-term exposure issues.

The fluorometry technique also has the potential to measure very small quantities of $NH_4^+$. Its original development was for measuring ppb-level concentrations in natural waters, so if extraction from silicates can be complete, there is no reason to think a similar level of precision could not be developed for geologic samples.

## 4.3 Suggestions for fluorometry improvement

The most difficult to quantify aspect of the fluorometry and colourimetric methods are the efficiency of the extraction of $NH_4^+$ during HF digestion. since HF digestion is only partial, assessing the amount of $NH_4^+$ that remains in undigested materials could prove valuable. Undigested materials are likely predominantly organics and/or oxides. Oxides should have low $NH_4^+$, as there are no crystallographic spaces in mineral lattices to accommodate $NH_4^+$. Organic content is typically low in crystalline rocks (e.g., granites, gneisses), but would contribute N to bulk rock concentrations. Minimizing the amount of rock sample powder used may increase the efficiency of extraction. Indeed, preliminary tests suggest that measured concentrations are not affected down to the initial rock powder mass of 0.1 g (Fig 6), though at very small sample sizes homogeneity issues could become apparent.



**Table 5.** Nitrogen from all three techniques, mass spectrometry, colourimetry, and fluorometry compared to published values from NAA. Shown are mean with standard deviation and coefficient of variation in parentheses. All concentrations are in ppm.

| Standard | Published | Colourimetric | Mass spectrometry | Post-distillation | Fluorometry |
|---|---|---|---|---|---|
| BCR-1/2 | $34 \pm 12$ (0.35) | $12.6 \pm 8$ (0.63) | 21 | $38 \pm 1$ (0.02) | $33 \pm 8.3$ (0.25) |
| BHVO-2 | $22.6 \pm 3$ (0.13) | $3.5 \pm 0.7$ (0.2) | $13.3 \pm 0.6$ (0.05) | $35 \pm 1$ (0.03) | $15 \pm 5.7$ (0.38) |
| G-2 | $34 \pm 4$ (0.12) | $1.6 \pm 0.9$ (0.56) | $5 \pm 0.7$ (0.14) | $36 \pm 4$ (0.11) | $11 \pm 4.9$ (0.45) |
| SY-4 | | $6.9 \pm 2.8$ (0.41) | $14.3 \pm 0.6$ (0.04) | $11 \pm 1$ (0.09) | $5.2 \pm 4.5$ (0.87) |
| LKSD-4 | | $487 \pm 401$ (0.82) | $16000 \pm 8$ (<0.01) | $9300 \pm 2150$ (0.23) | $5200 \pm 1000$ (0.19) |
| Till-4 | | $82.2 \pm 40$ (0.49) | $440 \pm 2$ (<0.01) | $455 \pm 36$ (0.08) | $71 \pm 25$ (0.35) |
| Carb | | 66.1 | $48.5 \pm 1.3$ (0.03) | 37 | $93 \pm 18$ (0.19) |

There are other small areas for fluorometry improvement. One is to filter samples after neutralizing with HF, as sediment may affect fluorescence readings. The second is to attempt a neutralizing agent other than KOH, which may contain trace levels of $NH_4^+$. KOH was used initially due to its better performance (Hall, 1993) during distillation.

As noted by Ader et al. (2016), acquisition and development of a robust international standard would be extremely helpful
5   in this or any future analytic technique development. Our work demonstrates that existing rock standards may be suited to this charge.

### 4.4 Preliminary application - continental crust

As a demonstration of the potential of the method, we calculated a N budget of the continental crust using analyses of a variety rock types. Most samples have come from Canada, though several are from other areas of North America. We present only
10   averages here, full analyses are available in the supplemental material.

We calculate the N content of the upper continental crust using two approaches. The first is to use glacial tills as an average of upper crustal rocks (Tables 4, 7). Again, this should be regarded as a proof of concept approach, as these tills are all from British Columbia, Canada, and primarily erode Phanerozoic rocks. The second method relies on rock abundance data after Wedepohl (1995), measured N in those rocks either from this study (Tables 4, 7) or as compiled in Johnson and Goldblatt
15   (2015).

The two upper crust approaches yield results that are distinct from each other (Tables 4, 8). The rock abundance approach is most similar to Johnson and Goldblatt (2015). A wider range of till samples, which have eroded a greater variety of crustal ages and rock types, could address this discrepancy.

In addition, we use xenolith data to approximate the mafic lower crust and gneisses to approximate the felsic lower crust.
20   These results are quite preliminary, and further analysis of lower crustal samples would assist interpretation greatly. By combin-



**Figure 6.** Measured concentration for rock standards normalized to mean concentration from analysis day plotted against the mass rock powder. Symbols represent standards, and all analyses shown are fluorometry with and without distillation. Decreasing initial sample mass does not seem to affect concentrations calculated, as measured concentrations show no significant trend away from the mean value given different amounts of rock powder dissolved.



**Table 6.** Comparison of different method requirements and performance. SPAD stands for samples per analysis day, and indicates approximate number of samples that can be run in one day, after all prep work has been completed. Accessibility is a qualitative measure of how common analytical equipment is in the geochemical community. Reproducibility is the average of the coefficient of variation from Table 5. Sensitivity indicates minimum concentration able to be potentially measurable.

| | Colourimetric | EA-Mass spectrometry | Fluorometry | NAA |
|---|---|---|---|---|
| Prep time required | 1 week digestion | 1 week | 1 week digestion | weeks to months |
| SPAD | $< 30$ | 18 | 50, or $< 30$ with distillation | |
| Species measured | $NH_4^+$ | easily combusted organic, some $NH_4^+$ | $NH_4^+$ | total |
| Accessibility of equipment | high | medium | high | low |
| Reproducibility | 0.52 | 0.05 | 0.38, 0.09-distilled | 0.2 |
| Sensitivity | $\sim 5$ ppm | $\sim 10$s ppm | ppb | ppm |
| Reagent toxicity | HF-high | low | HF-high | radioactivity |

ing lower crustal our estimate with upper crust estimates, we find total crust N to be $1.1 \pm 0.4 \times 10^{18}$ kg using the till+xenolith approximation and $1.6 \pm 0.6 \times 10^{18}$ kg using rock proportion+xenolith. While these are the same within error, we note that this is primarily due to "diluting" the upper crust estimates with lower crustal values.

   As an exercise to assess the impact of uncertainty in fluorometry measurements, we can perform the same budget calculations
by using the highest and lowest analyzed values of BCR-2 compared to the mean value to bracket calculation accuracy. Given a maximum measured value of 42 ppm and a minimum measured value of 21 ppm (Supplemental), compared to a mean of 33 ppm for BCR-2, we can assess the effect of multiplying continental budgets by 1.3 and 0.64. This leads to a till-based upper crust range of $0.5 \times 10^{18}$ to $1.1 \times 10^{18}$ kg N, rock-abundance based upper crust range of 0.8 to $1.7 \times 10^{18}$ kg N, and lower crust of 0.2 to $0.4 \times 10^{18}$ kg N. These differences in continental N budget estimates do not change the broad agreement of our
proof of concept budget with previously published work, that the continental crust contains $\sim 0.5$ present atmospheric mass of N (PAN = $4 \times 10^{18}$ kg N). For large scale questions, fluorometry is an appropriate technique. For questions that require finer resolution, such as biologic incorporation of rock-bound N (Morford et al., 2011), more method development is required.

   The fluorometry technique has the potential to increase the number of analyses of under-sampled rock types such as volcanics and middle to lower crustal xenoliths. These poorly sampled reservoirs have the potential to sequester large amounts of N, and
15 the continental crust can be both a long-term storage reservoir of N and an important source of biologically available N (Morford et al., 2011). Thus, determining its abundance is of interest to both geology and biology.



**Table 7.** Nitrogen concentration in upper and lower crustal rocks based on Wedepohl (1995). Proportions are of upper or lower crust mass, N concentration (ppm) are from this study (where error is shown) or from Johnson and Goldblatt (2015). Nitrogen contribution is simply concentration multiplied by proportion of crust. We use gneisses as a proxy for felsic granulites, and xenoliths for mafic granulites. Values from Johnson and Goldblatt, 2015 (JandG) given for comparison, with error shown as standard error of the mean.

| Rock type | Proportion of crust (%) | N (ppm) | N contribution | JandG |
|---|---|---|---|---|
| **Upper crust** | | | | |
| Shale/silt | 6.16 | $1064 \pm 113$ | $65.6 \pm 7.0$ | $860 \pm 64$ |
| Sandstone | 2.94 | 230 | 6.8 | $230 \pm 110$ |
| Volcanics | 2.80 | $21. \pm 12.54$ | $0.6 \pm 0.3$ | $50 \pm 60$ |
| Carbonates | 1.96 | $114.2 \pm 40.9$ | $2.2 \pm 0.8$ | $130 \pm 17$ |
| Granitic | 45 | $30.8 \pm 31.2$ | $13.8 \pm 14.0$ | $54 \pm 7$ |
| Tonalite | 5 | 24 | 1.2 | $24 \pm 7$ |
| Gabbro | 6 | $11.3 \pm 12.6$ | $0.7 \pm 0.8$ | $5 \pm 2$ |
| Gneisses | 19.20 | $29.8 \pm 0.8$ | $5.7 \pm 0.2$ | $135 \pm 50$ |
| Mica schist | 4.80 | 500 | 24.0 | $500 \pm 44$ |
| Amphibolites | 5.40 | 22 | 1.2 | $22 \pm 10$ |
| Marble | 1 | 1000 | 9.0 | $1000 \pm 500$ |
| **Total Average** | | | $131 \pm 14.1$ | $150 \pm 12$ |
| **Lower crust** | | | | |
| Felsic granulites | 62 | $29.8 \pm 0.8$ | $18.4 \pm 0.5$ | $17 \pm 6$ |
| Mafic granulites | 38 | $34 \pm 16.1$ | $13.1 \pm 6.1$ | $17 \pm 6$ |
| **Total Average** | | | $31.5 \pm 3.1$ | $17 \pm 6$ |

## 5 Conclusions

We have measured N concentration in a number of rock standards using three different methods: EA-mass spectrometry, colourimetry, and newly adapted fluorometry, and compared them to previously published values using neutron-activation analysis. Our analysis shows that fluorometry reproduces previously published values for BCR-2, and may also do so for

5 BHVO-2 and G-2 given an additional distillation step. Fluorometry appears more well-suited to measuring geologic $NH_4^+$ in silicate rocks than either colourimetry or EA-mass spectrometry, while mass spectrometry is more well suited to high-N samples with significant organic N. No one method appears to be a "gold standard" for geologic N analysis, and we call for further development in this area. There are several suggested avenues for improving fluorometry, namely improving HF-



**Table 8.** Total continental crust N based on tills, rock proportions, and xenolith concentrations. Our results are consistent with previous work that suggests there is about $2 \times 10^{18}$ kg N in the continental crust. All N masses are $10^{18}$ kg.

| Reservoir | Mass in kg (% of total) | N (ppm) | N mass |
|---|---|---|---|
| Upper crust (tills) | $1.01 \times 10^{22}$ (53%) | $81.2 \pm 36.4$ | $0.82 \pm 0.4$ |
| Upper crust (rock abundance) | $1.01 \times 10^{22}$ (53%) | $131 \pm 14.1$ | $1.32 \pm 0.1$ |
| Lower crust | $8.9 \times 10^{21}$ (47%) | $34.4 \pm 16.1$ | $0.31 \pm 0.03$ |
| Total crust (Till + lower) | $1.9 \times 10^{22}$ | $58 \pm 21$ | $1.1 \pm 0.4$ |
| Total crust (abundance + lower) | $1.9 \times 10^{22}$ | $84 \pm 11$ | $1.6 \pm 0.1$ |

digestion efficiency and fine tuning of HF-neutralization. Minimizing the volume of liquid required for digestion (HF) or neutralization would increase the sensitivity of the method, which could work for very low concentrations, down to ppb levels.

To demonstrate a potential application of fluorometry, we calculated a continental N budget. This budget is based on analysis of glacial tills (proxy for upper crust), a number of Phanerozoic volcanics, and a variety of mid-crustal xenoliths to augment

existing literature analyses. Our approach estimates that $1.1 \pm 0.4 \times 10^{18}$ kg N (till+xenoltih approach) and $1.6 \pm 0.6 \times 10^{18}$ kg N (rock abundance approach) is in the continental crust, consistent with recent estimates (Goldblatt et al., 2009; Johnson and Goldblatt, 2015). The fluorometry technique appears most appropriate for these large scale questions, where exact precision is not required. An additional application could be as an initial analysis to determine approximate concentration, which is a key step in further isotopic investigations.

All methods assessed herein have strengths and weaknesses, which are amplified due to the ability of N to exist as multiple species in the same sample. We call here again for the development of internationally accepted geologic N standards. New method development is difficult without such standards, and care should be taken to classify what species of N ($NO_3$, $NH_4^+$, organic N, $N_2$) is being measured. We also report the first $\delta^{15}N$ values for a series of rock standards, as isotope values should be part of any international standard development.

*Author contributions.* Benjamin W Johnson, Rana El-Sabaawi, and Colin Goldblatt designed the experiments and overall structure of the research. Samples were collected/procured by Benjamin W Johnson and Natashia Drage. Analysis was done by Benjamin W Johnson, Natashia Drage, and Nova Hanson. Laboratory support and technical input were given by Jody Spence. All authors contributed to the study.

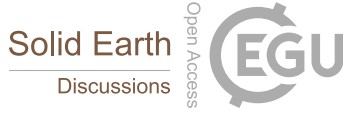



*Acknowledgements.* The authors would like to thank NSERC Discovery grant to CG for funding CG and BWJ, a University of Victoria NSERC Undergraduate Student Research Award to ND, and NSERC awards to RES for funding support. Nitrogen isotope measurements were done at the University of Washington, and funded by the Virtual Planetary Laboratory.



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
