# Peer review of "Measurement of geologic nitrogen using mass spectrometry, colourimetry, and a newly adapted fluorometry technique"

_Solid Earth, 2016_

## Referee Comment (RC1) · P. Barry (Referee) · 12 Dec 2016

Referee Comment of: Measurement of geologic nitrogen using mass spectrometry, colourimetry, and a newly adapted fluorometry technique, by Johnson and Coauthors.

In the manuscript the authors measured a number of geochemical rock standards using 3 different techniques: mass spectrometry, colourimetry, and fluorometry. The fluorometry approach is a novel adaptation of a technique commonly used in biologic science, and is applied in the paper to assess geologic $NH_4^+$. This work is important and informative for the geochemical community that is broadly interested in volatile cycling and N speciation in the Earth. The results presented here represent a valuable first step towards producing an internationally recognized standard for N in solid materials.

[Figure]

The discussion of the different techniques is also quite valuable, and the authors have thought outside of the box on how to make N measurements more efficiently. However, more emphasis should be placed on the utility of obtaining N isotopes, which is only possible using the mass spec technique. The results of the study are not fully satisfying (some lack of agreement), however they represent an important step forward and should be published in Solid Earth.

Comments:

Abstract: I agree that measuring NH4 is extremely important, especially for understanding the N composition of Earth's crust. The fluorometry technique sounds promising, but as the paper reveals has it's shortcomings as well (i.e., it's still a work in progress). Furthermore, the development of N standards for rocks is much welcomed and it would be good to determine the N isotopic ratio of such standards in many of the labs worldwide (i.e., Tokyo, Nancy, Scripps).

Introduction: The authors are correct to point out the difficulty of measuring N in rocks and that very few labs do this, thus I agree that a new and cheaper technique would greatly benefit the community at large. But it needs to be highlighted that these alternative techniques (colourimetry and fluorometry ) are also time consuming – requiring chemical digestions and irradiation. They also cannot be used to determine N isotopes, which in my opinion is a major shortcoming.

Line by line comments:

Pg 2 l 23 – does it also work for other species of N? What exactly is mean by 'straightforward'? Has this technique been compared with standard techniques to show that there is no fractionation?

Pg 4 l 83 – is this a step that the authors are taking (i.e., distributing the samples to be analyzed isotopically by the larger community in order to test the reproducibility of these standards)?

Pg 5 l 40 – you describe the distillation process in detail, but how much time is needed to complete this work? It seems like it will take a large amount of time (days to weeks) in which case it becomes more difficult to argue that this technique is more efficient than standard techniques.

L 58 – should this reference be in brackets?

Pg. 6 table 2 – why do the uncertainties on the isotopic values vary so much? From 0.1 to 1.3?

Table 4 – explicitly state that these are in units of ppm

Pg 7 figure 1 – can you explain exactly what absorbance means?

Pg 8 l 14 – but importantly the mass spec work will also provide invaluable isotopic information

Pg 10 figure 4 – how does this correction work over 0.35 KOH %?

Table 5 – can this table be combined with table 3? Seems to be much of the same info. . .

Pg 13 l 4 – a more detailed description of the rocks would be welcomed

Dr. Peter Barry peter.barry@earth.ox.ac.uk

---

## Referee Comment (RC2) · Anonymous Referee #2 · 21 Dec 2016

This paper compares several techniques to quantify ppm-level of nitrogen trapped in geological samples. Authors are particularly interested in crustal rocks, where a large portion of nitrogen is believed to be trapped in the ammonium form. As authors point out, understanding the behaviors of these type of nitrogen could lead to answering to the question how nitrogen cycled within the surface region of Earth that consist of atmosphere, biosphere and crusts, and also between the surface region and the mantle.

I agree with the most general authors' point that nitrogen study in various crustal rocks bears substantial scientific significance. However, I am not always convinced with the authors' strategy obtaining large numbers of plain concentration data for the ammo-

nium form nitrogen in crustal rocks, aiming at a novel understanding regarding the nitrogen behavior in the crusts. Nitrogen quantification in rocks itself has been done since several decades ago by several techniques. What kind of a new finding they expect by now? The example they introduce in 4.4 essentially says that they can roughly confirm the estimation of nitrogen budget in crustal rocks, which was already done several decades ago.

Many of the important references are missing in this paper.ãĂĂI don't request authors to do a thorough review of the previous techniques, but they should at least describe in the paper what is the standard techniques used for nitrogen study in rock samples. One of the technique is mass-spectrometry. Indeed, authors introduce one example of mass-spectrometry. However, the technique they refer to is not the one normally used for rock studies. Putting samples in a Tin-capsule and heating it to 1000oC is a technique to be applied to biological (easier-to-combust and with large N concentration) samples. See numbers of papers, for instance, by S. R. Boyd, S. E. Bebout, D. Haendel or D. L. Pinti, on ammonium form nitrogen trapped in sedimentary or crustal rocks. They all care for the nitrogen extraction problem or contamination issue, and present reasonable solutions. I understand well that authors would like to sell the fluorometry technique, but the comparison with other techniques must be done in a fair manner. The other technique I realized missing in this paper is the Kjeldahl method. This is a well-established chemical technique to extract and quantify ammonium nitrogen, therefore, must be directly compared with the two chemical techniques introduced in the paper. For example, Honma and Itihara, GCA (1981) measured numbers of crustal rocks by this method. Essentially what is the new selling point in the fluorometry technique? Isn't it just a variation of the classical Kjeldahl method?

In summary, I consider that references to other techniques are absolutely missing or poorly described in this paper, which prevents readers from reasonably understanding the pros and cons of the techniques.

Misc: 1. Briefly summarize the principle of techniques in the introduction, rather than

to just say read this paper. 2. Table 3. I don't understand why the concentration of ammonium "in the sample" change after the distillation process. Distillation means just vaporizing water, isn't it? Why does the differences of concentration before and after the distillation differ between samples (e.g., among BCR-1/2, BHVO-2 and G-2). I am a bit worried to find that the post-distillation concentrations are curiously similar among these three.

---

## Author Comment (AC1) · 11 Feb 2017

**Measurement of geologic nitrogen using mass spectrometry, colourimetry, and a newly adapted fluorometry technique**

We thank both reviewers for the comments, which were insightful and constructive. Our responses to reviewer comments (shown in grey) are given in black.

Reviewer 1: P. Barry

In the manuscript the authors measured a number of geochemical rock standards using 3 different techniques: mass spectrometry, colourimetry, and fluorometry. The fluorometry approach is a novel adaptation of a technique commonly used in biologic science, and is applied in the paper to assess geologic NH4+. This work is important and informative for the geochemical community that is broadly interested in volatile cycling and N speciation in the Earth. The results presented here represent a valuable first step towards producing an internationally recognized standard for N in solid materials.

The discussion of the different techniques is also quite valuable, and the authors have thought outside of the box on how to make N measurements more efficiently. However, more emphasis should be placed on the utility of obtaining N isotopes which is only possible using the mass spec technique. The results of the study are not fully satisfying (some lack of agreement), however they represent an important step forward and should be published in Solid Earth.

Comments:
Abstract: I agree that measuring NH4 is extremely important, especially for understanding the N composition of Earth's crust. The fluorometry technique sounds promising, but as the paper reveals has it's shortcomings as well (i.e., it's still a work in progress). Furthermore, the development of N standards for rocks is much welcomed and it would be good to determine the N isotopic ratio of such standards in many of the labs world-wide (i.e., Tokyo, Nancy, Scripps).
We certainly agree that the technique remains a work in progress. We also agree that sending our standards out to various labs would be quite useful and informative. While beyond the scope of this paper, this suggestion is welcome and we will use it as a starting point for a discussion with the geologic N community at large.

Introduction: The authors are correct to point out the difficulty of measuring N in rocks and that very few labs do this, thus I agree that a new and cheaper technique would greatly benefit the community at large. But it needs to be highlighted that these alternative techniques (colourimetry and fluorometry) are also time consuming – requiring chemical digestions and irradiation. They also cannot be used to determine N isotopes, which in my opinion is a major shortcoming.
The inability to measure isotopes is a drawback of the method, and it also requires a period of sample digestions. We do suggest, though, that sample throughput is still high, as many samples can be dissolved at once and processed on a single analysis day. Specifically, in lines 26-28 in the introduction we write:

> We emphasize, however, that N-isotopes cannot be measured with fluorometry, and both fluorometry and colourimetry techniques require a period of rock digestion which other techniques may not require.

Line by line comments:
Pg 2|23 – a) does it also work for other species of N? b) What exactly is mean by 'straight-forward'? c) Has this technique been compared with standard techniques to show there is no fractionation?
a) The OPA reagent is sensitive to amino acids, but the addition of sodium sulfite eliminates this sensitivity, rendering it active with NH4+ only. Without sodium sulfite, the method could perhaps be

sensitive to organic N in amino acids. A difficulty, however, is that HF dissolution will not liberate organic matter, and a different dissolution technique would be required. We have added a paragraph in section 4.3 addressing this as a possible avenue for improvement.

b) By straight-forward we mean the reagents and analytical procedure are simple. That is, there are only a few steps, and the equipment needed is not overwhelming. We now say:

> The fluorometry technique has the advantage over other techniques by being relatively fast, requiring few reagents, requiring more accessible analytical equipment, as well as specifically targeting NH4+.

c) Are you referring to isotopic fractionation? If so, as far as we are aware, this is the first study to utilize the fluorometric technique in this way, and we cannot measure isotopes using fluorometry. If you are referring to perhaps conversion of NH4+ to NH3, NO3, or N2 during the analysis, we have taken care to maintain a low pH (which favours NH4 over NH3) and limit the time sample solutions sit before analysis to avoid conversion to another species.

Pg 4| 83 is this a step that the authors are taking (i.e., distributing the samples to be analyzed isotopically by the larger community in order to test the reproducibility of these standards)?
This suggestion is beyond the current scope of the paper, but we are using your comment as an impetus to begin this discussion with other members of the geological N-isotope community.

Pg 5|40 – you describe the distillation process in detail, but how much time is needed to complete this work? It seems like it will take a large amount of time (days to weeks) in which case it becomes more difficult to argue that this technique is more efficient than standard techniques.
You are correct in noting the addition of considerable time if the distillation step is taken. Specifically, we note in Section 4.2, paragraph 2:

> Distillation takes 15-20 minutes per sample, limiting throughput, and makes either fluorometry or colourimetry more on par with mass spectrometry in terms of time needed for analyses. Additionally, it is difficult to consistently distill the same volume for each sample, which limits accuracy and reproducibility.

L 58 – should this reference be in brackets?
Yes, thank you.

Pg. 6 table 2 – why do the uncertainties on the isotopic values vary so much? From 0.1 to 1.3?
Samples with low N concentration have larger uncertainties, likely due to contribution from the blank during analysis. Additionally, the crystalline samples (G-2, BHVO-2, BCR-2, and SY-4) likely contain most of their N as NH4+, which may not release uniformly during EA-combustion.

Table 4 – explicitly state that these are in units of ppm
Thank you, fixed.

Pg7 figure 1 – can you explain exactly what absorbance means?
We now say in this figure caption:

> "Absorbance is the difference between light that enters the sample cuvette and the light that transmits through the sample cuvette to the detector. Values larger than 1 are due to dilution corrections."

Pg 8|14 – but importantly the mass spec work will also provide invaluable isotopic information
Agreed. We make this explicit in section 4.2:

Mass spectrometry has the major advantage over fluorometry or colourimetry by being able to measure N-isotopes in a given sample. Isotopic values are crucial in determining N-cycling, both biologically and in geologic reservoirs. One application of the fluorometry technique is as a ``first-pass'' analysis to determine N concentration. The concentration of N in a sample dictates what type of mass spectrometric technique (e.g., EA, off-line combustion, etc.) is most appropriate for isotopic analysis.

All analyses shown in this study were done at KOH concentrations between 0.5 and 0.2% KOH. For KOH concentrations between 0.2 and 0.35%, one could make a linear correction. Corrections at concentrations higher than 0.35% KOH would be difficult, and we recommend avoiding these high concentrations of KOH.

Table 5 -can this table be combined with table 3? Seems to be much of the same info....
Our thought behind separating these tables was to provide some more detail (i.e., number of analyses) regarding the fluorometric analyses.

We have now added number of analyses to what was Table 5, now Table 4, as we have removed Table 3.

We have added a supplemental text file with lithologic descriptions, as well as more detail on other references that have analyzed these samples. In addition, we have added a read-me text file to the supplemental information to assist with accessibility of the data files.

**Anonymous Referee #2**
This paper compares several techniques to quantify ppm-level of nitrogen trappe in geological samples. Authors are particularly interested in crustal rocks, where a large portion of nitrogen is believed to be trapped in the ammonium form. As authors point out, understanding the behaviours of these type of nitrogen could lead to answering to the question how nitrogen cycled within the surface region of Earth that consist of atmosphere, biosphere and crusts, and also between the surface region and the mantle.

I agree with the most general authors' point that nitrogen study in various crustal rocks bears substantial scientific significance. However, I am not always convinced with the authors' strategy obtaining large numbers of plain concentration data for the ammonium for nitrogen in crustal rocks, aiming at a novel understanding regarding the nitrogen behaviour in the crusts. Nitrogen quantification in rocks itself has been done since several decades ago by several techniques. What kind of a new finding they expect by now? The example they introduce in 4.4 essentially says that they can roughly confirm the estimation of nitrogen budget in crustal rocks, which was already done several decades ago.
The preliminary application to the continental crust was meant to serve both as a check on the capability of the method and to add analyses of poorly measured continental rocks. We stress that we used our measurements to augment existing measurements, and also to use tills as a proxy for upper continental crustal composition. The latter has not been attempted for N, though it has been used for many other elements (Gasching et al., 2016). In addition, estimates of crustal N content have varied substantially over the past 40 years, ranging from the $<1\times10^{18}$ kg N (Rudnick and Gao, 2014) to $14\times10^{18}$ kg N (Delwiche, 1970), with the most current estimate being $\sim2\times10^{18}$ kg N. The atmosphere contains $4\times10^{18}$ kg N, so this range in estimates is substantial compared to the mass of the atmosphere.

As our technique is able to reproduce existing estimates of continental crust N content, we suggest then that it is well-suited to measure many other poorly characterized geologic reservoirs.

Many of the important references are missing in this paper. I don't request authors to do a thorough review of the previous techniques, but they should at least describe in the paper what is the standard techniques used for nitrogen study in rock samples. One of the technique is mass-spectrometry. Indeed, authors introduce on example of mass-spectrometry. However, the technique the refer to is not the one normally used for rock studies. Putting samples in a Tin-capsule and heating it to 1000oC is a technique to be applied to biological (easier-to-combust and with large N concentration) samples. See numbers of papers, for instance by S.R. Boyd, S.E. Bebout, D. Haendel or D.L. Pinti, on ammonium for nitrogen trapped in sedimentary or crustal rocks. They all care for the nitrogen extraction problem or contamination issue, and present reasonable solutions. I understand well that authors would like to sell the fluorometry technique, but the comparison with other techniques must be done in a fair manner. The other technique I realized missing in this paper is the Kjeldahl method. This is a well-established chemical technique to extract and quantify ammonium nitrogen, therefore, must be directly compared with the two chemical techniques introduced in the paper. For example, Honma and Itihara, GCA (1981) measured numbers of crustal rocks by this method. Essentially what is the new selling point in the fluorometry technique? Isn't it just a variation of the classical Kjeldahl method?

In summary, I consider that references to other techniques are absolutely missing or poorly described in this paper, which prevents readers from reasonably understanding the pros and cons of the techniques. Misc: 1. Briefly summarize the principle of techniques in the introduction, rather than to just say read this paper.

We have added a new paragraph in the introduction to provide discussion of these methods. We stress that we chose the EA-combustion technique because this is the most commonly used method to measure N. You are correct to point out that EA-combustion in a tin capsule is not the most appropriate for mineral-bound N (Brauer and Hahne, 2005). The other techniques you have highlighted here are certainly appropriate for measuring mineral-bound $NH_4^+$, but they do require dedicated preparation lines or aggressive dissolution techniques (high-pressure bombs). Our goal with the fluorometry is not to supplant these, necessarily, but rather to offer a more straight-forward alternative that requires non-high pressure or high-temperature extraction techniques.

2. Table 3 – I don't understand what the concentration of ammonium "in the sample" change after the distillation process. Distillation means just vaporizing water, isn't it? Why does the differences of concentration before and after the distillation differ between samples (e.g., among BCR-1/2, BHVO-2, and G-2). I am a bit worried to find that the post-distillation concentrations are curiously similar among these three.

Note that this data, originally in Table 3, has been moved to Table 4.

The distillation process preferentially removes $NH_4$ from the KOH-neutralized solution in addition to water. Distillation will also serve to separate $NH_4$ from any particulate matter in the sample, which could improve fluorometric accuracy, since particulates could interfere with the fluorometer. While the three post-distillation samples (BCR-1/2, BHVO-2, and G-2) are similar in concentration, they also match previously published values based on neutron activation analysis. In addition, repeated measurements of BCR-2 done without distillation are equivalent to to those post-distillation, indicating that for basaltic samples distillation is not required.